

# Uneven distribution of enamel in the tooth crown of a Plains Zebra (*Equus quagga*)

Daniela E. Winkler and Thomas M. Kaiser

Center of Natural History (CeNak), University of Hamburg, Hamburg, Germany

## ABSTRACT

Unworn teeth of herbivorous mammals are not immediately functional. They have to be partially worn to expose enamel ridges which can then act as shear-cutting blades to break the food down. We use the Plains Zebra (*Equus quagga*) as a hypsodont, herbivorous model organism to investigate how initial wear of the tooth crown is controlled by underlying structures. We find that the enamel proportion is smaller at the apical half of the tooth crown in all upper tooth positions and suggest that lower enamel content here could promote early wear. Besides this uneven enamel distribution, we note that the third molar has a higher overall enamel content than any other tooth position. The M3 is thus likely to have a slightly different functional trait in mastication, resisting highest bite forces along the tooth row and maintaining functionality when anterior teeth are already worn down.

## INTRODUCTION

Hypsodonty has been defined as the relative increase in the height of cheek teeth (*Van Valen, 1960*), or more simply as larger tooth crown height compared to tooth crown length (*White, 1959*; *Thenius, 1989*; compare Fig. 1). It is a common evolutionary strategy of herbivorous mammals to counter high abrasive loads in the ingested diet, which result in a high degree of dental wear. Following *White (1959)*, cusp hypsodonty, tooth base hypsodonty and root hypsodonty can be distinguished, depending upon which structure is elongated. Hypsodonty can be easily achieved in all tooth positions by extending specific ontogenetic phases during tooth development (*von Koenigswald, 2011*). Newly erupted hypsodont cheek teeth share a feature among all taxa: they are not immediately functional. To comminute tough plant matter, the relatively rounded apex of the (pre)molar tooth crowns has to wear down slightly, exposing the enamel ridges which may then act as shearing blades during mastication. The rapid wear of the topmost tooth crown has been noted in selenodont molars (*Osborn & Lumsden, 1978*), and several authors have hypothesised how this initial wear is facilitated. One theory is that empty chewing movements (thegosis) sharpen teeth in adults and initiate wear in young animals (*Every, 1972*; *Teaford & Walker, 1983*; *Every, Tunnicliffe & Every, 1998*). More often, however, such empty chewing is considered a behavioural anomaly (termed bruxism or pathological

Corresponding author
Daniela E. Winkler,
daniela.winkler@uni-hamburg.de

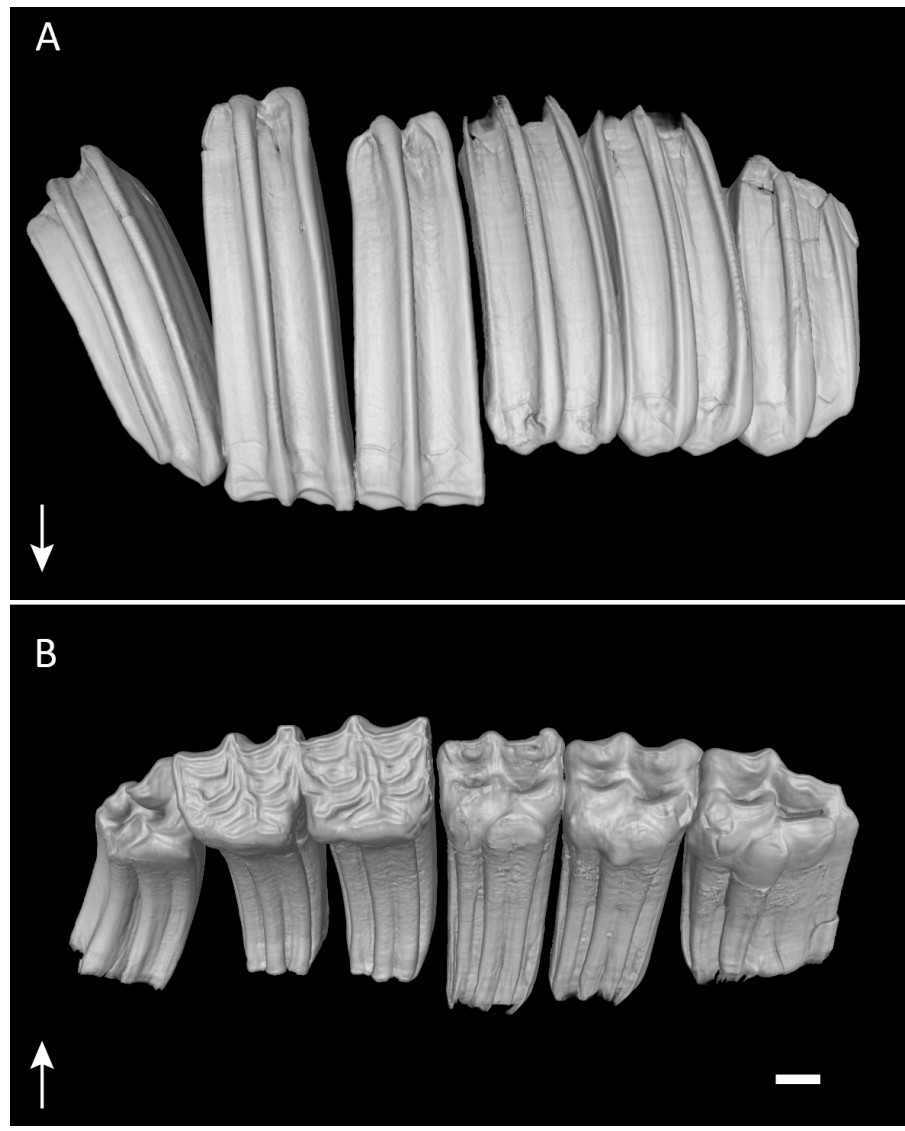

**Figure 1 Permanent dentition of *Equus quagga*.** Virtual 3D-model of the permanent upper right dentition of *Equus quagga*, no. 70335. As typical in hypsodont species, the tooth crowns are much longer than wide. Arrows point towards apical. (A) Anatomical position. Note that the caps of the deciduous premolars have been lost, but would have been present at this life stage. (B) Occlusal view. Scale bar 1 cm.

thegosis) which appears in livestock, other domestic or captive animals (e.g., *Murray & Sanson, 1998*; *Troxler, 2007*; *Troxler, 2012*) and also in man (*Manfredini et al., 2013*). We do not think that thegosis is needed to initiate functionality and a lack of empirical data are available in support of this hypothesis.

We therefore propose that the apex of the tooth crown should be less resistant to all wear mechanisms. Abrasion (tooth-to-food contacts including adhesive particles such as grit and dust) and attrition (tooth-to-tooth contact during chewing movements) would hence be sufficient to promote early wear and expose functional enamel ridges quickly. This could be accomplished by either building the apex of the tooth crown from less and/or

thinner enamel or by building a less resistant enamel microstructure. Both hypotheses suggest that the apex of the tooth crown is structurally different from the rest of the tooth. Analysis of enamel microstructure at different tooth crown heights is a destructive and time consuming method, therefore we chose to study enamel distribution within the tooth crown of a subadult Plains Zebra (*Equus quagga* sp.) using micro CT-scanning. The Plains Zebra is an ideal model organism for a large, hypsodont herbivore, because it is adapted to grazing in both arid and savannah climates and therefore needs to have a high tolerance of abrasional tooth wear. Its diet is composed of up to 90% grass but also includes some browse (*Nowak, 1999*). Grass lamina is generally narrower and more uniform in thickness than browse. Vascular bundles are evenly distributed and generally very narrow. The fibre bundles in grasses are more evenly distributed and in higher levels than in most dicotyledonous browse. This requires more energy to separate the grass into smaller fragments and this is most economically achieved by shear-cutting, rather than crushing or grinding (*Archer & Sanson, 2002*). Grasses have higher internal abrasive contents than browse (*McNaughton et al., 1985*; *Piperno et al., 2002*) and often also bear more airborne grit and dust because they are common in arid, savannah-like habitats. Specialised grazing species therefore tend to be the most hypsodont, as they experience high abrasional wear. Amongst extant large herbivore species, the Equidae exhibit the highest degree of hypsodonty, only equalled by a few Bovidae like *Bison bison* (compare hypsodonty indices in *Janis, 1988*).

## MATERIAL AND METHODS

The selected individual (specimen no. 70335) is a loan from Museum für Naturkunde (Berlin). It shows very low or no wear on the premolar and molar teeth and is therefore in the optimal stage to investigate enamel distribution within all tooth positions of the same individual. The tooth eruption sequence for upper permanent teeth in *Equus quagga* is M1, M2, I1, P2, P3, P4, I2, C, M3, I3 (*Erz, 1964*). Hence, we see small amounts of material loss in the earlier erupting teeth M1 and M2 compared to the unworn P2, P3, P4 and M3. Before scanning the caps of the deciduous premolars must have been present, but were lost. The roots of all premolars and molars seem to be open, however, some parts of the roots might have been damaged during extraction from the skull. For this pilot study, we did not use unworn premolars and molars of several individuals, but one postcanine dentition of a single specimen. We focus on the upper permanent dentition, because upper teeth are employed as the standard in studying dental characteristics (*Archer & Sanson, 2002*; *Fortelius & Solounias, 2000*; *Solounias & Semprebon, 2002*) and functional traits should be more pronounced as compared to lower teeth (*Kaiser & Fortelius, 2003*) due to the lack of gravity impact. High resolution computed tomography (microCT) scans with a voxel size between 0.075 and 0.1 mm (see Appendix S1) were obtained at Steinmann-Institut für Geologie, Mineralogie und Paläontologie (Universität Bonn, Germany) on the CT scanner *v*| tome| × *s* (GE phoenix | x-ray). All teeth were scanned individually after extraction from the skull. The software VG StudioMax 2.1 (Volume Graphics, Heidelberg, Germany) was used for reconstruction of virtual models and further processing. First, each tooth

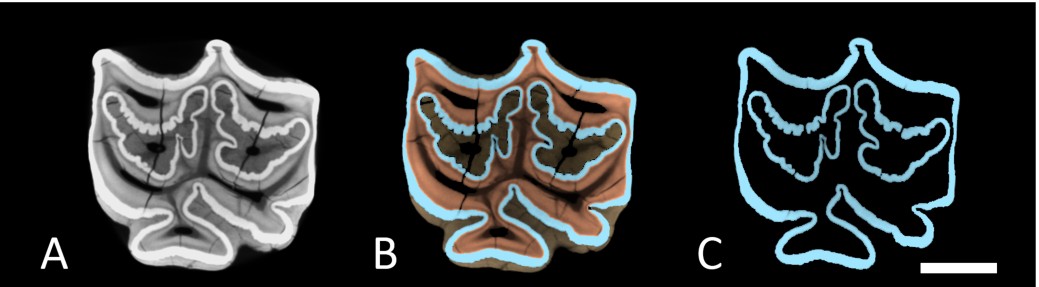

**Figure 2 Cross-sections through M1.** Cross section trough a virtual 3D-model of M1. (A) Full tooth model with all dental tissues. (B) Segmented tooth model with enamel in blue, dentin in orange and cementum in brown. (C) The same enamel only. Scale bar 1 cm.

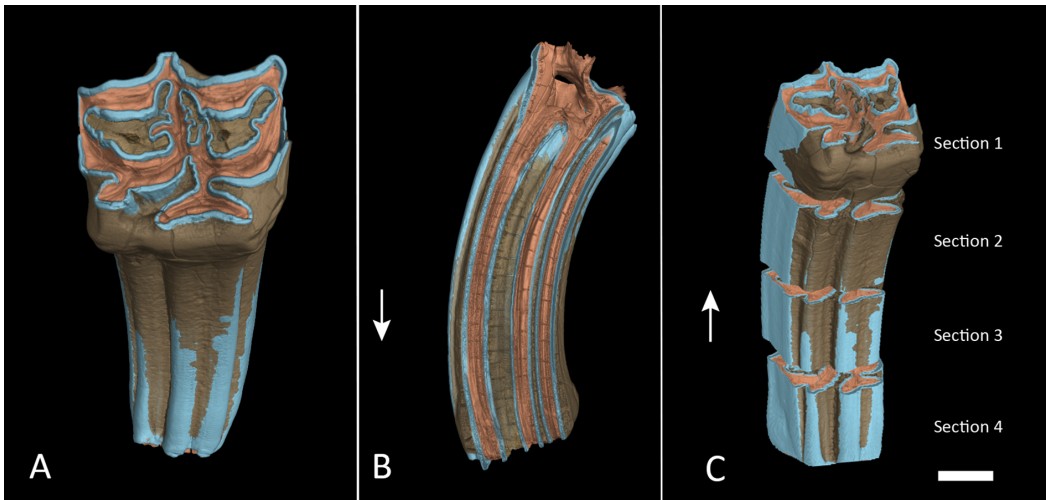

**Figure 3 Segmented M1 and sections.** (A) Segmented tooth model of M1. (B) Vertical cross section trough M1 in anatomical orientation with open roots. (C) The same as in (A) but with each section separated. The lower boundary of section 4 was cut approximately at the cementoenamel junction. Arrows point towards apical. Colours as in Fig. 2. Scale bar 1 cm.

was reconstructed with all dental tissues (enamel, dentin and cementum) as a voxel model using manual and automatic segmentation tools. All models were cut at the base of the crown by fitting a clipping plane through the cementoenamel junction. Then, only the voxels above the clipping plane were selected to create models of the tooth crown. Next the dental tissues enamel, dentin and cementum were individually segmented (Figs. 2A–2C, 3A and 3B).

We created a pure enamel voxel model and two full tooth models, one with all dental tissues, one without cementum, and cut them at approximately 75%, 50% and 25% of the initial crown height and created individual models of four tooth sections: Section 1 from the apex at 100–75% crown height, Section 2 from 75–50% crown height, Section 3 from 50–25% crown height and Section 4 from 25% down to the base of the crown (Fig. 3C). Cutting planes had to be inserted manually and could therefore not exactly represent quarters of the tooth crowns. Volumes of the enamel sections and full tooth sections were

taken directly from the properties of these models as displayed in VG StudioMax. We then calculated relative enamel content per section by dividing the enamel volume of that section by total volume of the respective section. We further subtracted the volume of cementum and repeated the same calculation without cementum contributing to the total volume. Finally, the quotient from enamel by dentin was calculated to express the ratio between those two dental tissues.

## RESULTS

Data on enamel content in relation to the whole tooth volume and the whole volume without cementum are given in Table 1. Though distribution of enamel content per section was variable between teeth, it was consistently smallest in Section 1 (the most apical section) for all tooth positions when total volume was considered. In relation to the tooth model without cementum, however, either Section 1 or Section 2 had the lowest enamel proportions. This shifted ratio results from a higher cementum deposition at the apex of the tooth. For the full tooth model, Section 4 had the highest enamel proportions within each tooth position. When cementum was left out, either Section 3 or Section 4 had the highest relative enamel content (compare Table 1). It is notable that M3 was composed of more enamel than all other teeth (>7% higher relative enamel content than all other tooth positions). This is well expressed in the enamel/dentin-ratio (Fig. 4B). From P2 to M2, each section except P4 Section 4 contained less enamel then dentin. For the M3, however, this ratio reversed as each section contained more enamel than dentin.

## DISCUSSION

The results of this study support our hypothesis that the apex of the tooth crown is structurally different from the rest of the tooth. We have shown that the overall enamel content is lowest at the crown apex and highest in the lower half of the crown, regardless of the influence of cementum.

There are relatively more soft dental tissues (dentin and cementum) at the apex of the crown and therefore this part of the tooth is prone to faster wear. We further note that the base of each tooth seems to be structurally "enhanced", as the larger enamel content should strengthen it and help resist high pressure and stress loads.

In *Equus quagga*, we find the third upper molar to be structurally different from all other upper teeth, as it has the highest proportion of enamel along the tooth crown. We relate this result to adaptive pressures related to two phenomena:

1. Mechanical constraint: As the M3 is closest to the temporomandibular joint, the highest masticatory forces can be generated here (*Greaves, 2012*). The high enamel content will then prevent excessive wear and maintain evenly distributed chewing forces.
2. Biogenetic constraints: The M3 is the last tooth to erupt in most mammals, and also in the Zebra. Therefore it is also the tooth position maintaining function when anterior teeth are already wearing out.

In general, by being more resistant to wear, M3 can thus compensate for the functional loss of anterior teeth. A similar observation was also made by *Kaiser (2002)* for the extinct

Winkler and Kaiser (2015), *PeerJ*, DOI 10.7717/peerj.1002

**Table 1 Enamel proportions with and without cementum.** Enamel proportions calculated with the whole tooth volume (enamel, cementum, dentin) and for the whole tooth without cementum.

| Stage | P2 | | P3 | | P4 | | M1 | | M2 | | M3 | |
|---|---|---|---|---|---|---|---|---|---|---|---|---|
| | Enamel proportion whole tooth | Enamel proportion without cementum | Enamel proportion whole tooth | Enamel proportion without cementum | Enamel proportion whole tooth | Enamel proportion without cementum | Enamel proportion whole tooth | Enamel proportion without cementum | Enamel proportion whole tooth | Enamel proportion without cementum | Enamel proportion whole tooth | Enamel proportion without cementum |
| 1 | 27.34% | 37.58% | 23.17% | 35.23% | 27.23% | 39.98% | 32.86% | 48.28% | 29.22% | 46.50% | 37.96% | 53.61% |
| 2 | 29.64% | 37.63% | 27.97% | 38.99% | 29.08% | 38.47% | 35.09% | 45.62% | 32.15% | 44.62% | 39.22% | 52.72% |
| 3 | 34.50% | 41.89% | 34.80% | 46.69% | 33.17% | 42.19% | 38.45% | 48.45% | 35.52% | 47.31% | 43.20% | 57.01% |
| 4 | 41.61% | 47.11% | 36.43% | 46.99% | 41.62% | 54.64% | 42.35% | 47.31% | 40.55% | 50.56% | 44.37% | 54.72% |
| Total | 31.69% | 40.17% | 28.75% | 40.39% | 31.33% | 42.64% | 36.91% | 47.63% | 34.00% | 47.14% | 41.34% | 54.62% |

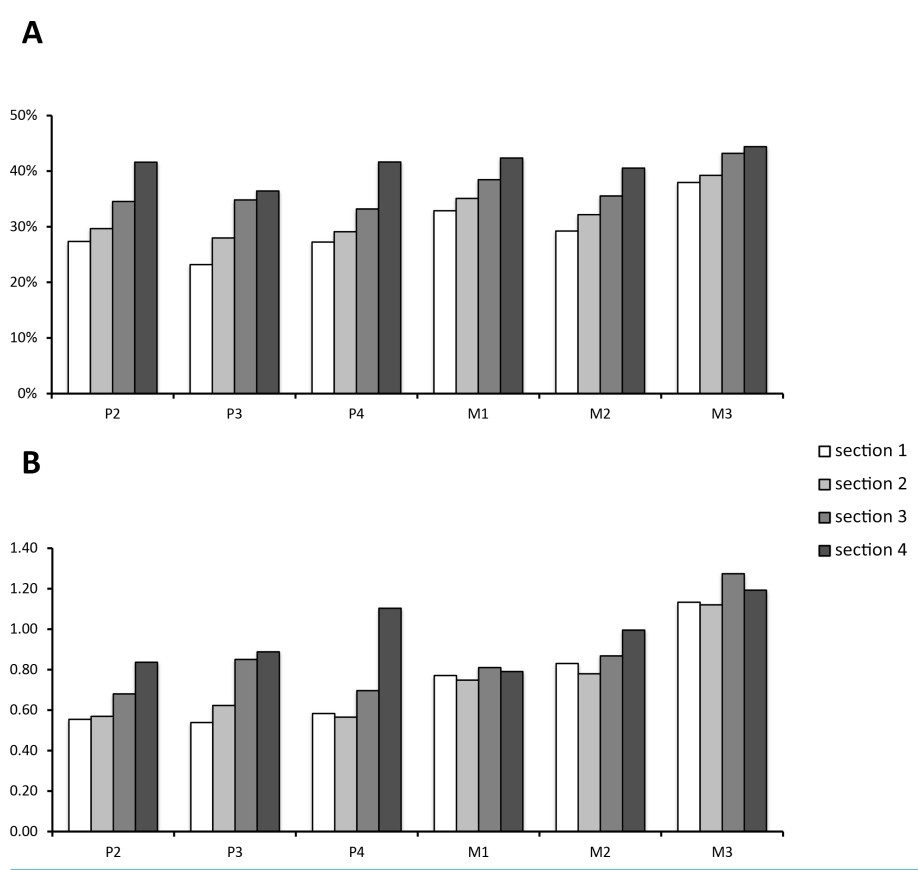

**Figure 4  Enamel content per section and tooth position.** (A) Relative enamel content per section and tooth position. Proportions are calculated using the whole tooth volume with all dental tissues (enamel, cementum, dentin). (B) The value of enamel versus dentin expressed as the quotient from enamel/dentin per section and tooth position.

equid *Cormohipparion occidentale*, in which M3 shearing function is optimised in later ontogenetic stages. We observe that the apex of the M3 crown also has the lowest enamel proportion compared to the remainder of tooth; however, the enamel content is higher than in the apex of any other tooth along the dentition. We hypothesise that it therefore takes longer for enamel ridges on M3 to become exposed and that wear is hence delayed, which further adds to M3 serving as a reserve once anterior teeth wear out.

Though *Equus quagga* is an appropriate model organism, these observations are still restricted to one single specimen of this taxon. They can, however, help us to refine hypotheses on how mechanical and ontogenetic constraints of wear and resistance may be solved in a biological system, by slight modification of a common structure. The findings suggest that different functional requirements at different tooth positions and tooth wear stages have shaped tooth morphology to a large extent and that even hypsodont forms are prone to high levels of functional adaptation. As mechanical constraints triggering these adaptations must universally apply to all mammals feeding on abrasive diets, we expect to find similar traits in other herbivorous species, including bovids.

## ACKNOWLEDGEMENTS

We thank our colleagues at Steinmann Institut for CT scanning of the specimen, the Museum für Naturkunde Berlin for specimen loan and Lucy A. Taylor (University of Oxford) for her suggestions to improve the language. We highly acknowledge the contribution of the three reviewers to the improvement of content and language of the manuscript. This research is publication no. 76 of the DFG Research Unit 771 "Function and performance enhancement in the mammalian dentition—phylogenetic and ontogenetic impact on the masticatory apparatus".

### Funding

This research was supported by the "Deutsche Forschungsgemeinschaft" (DFG, German Research Foundation, KA 1525/8-1). The funders had no role in study design, data collection and analysis, decision to publish, or preparation of the manuscript.

### Grant Disclosures

The following grant information was disclosed by the authors:
DFG, German Research Foundation: KA 1525/8-1.

### Competing Interests

The authors declare there are no competing interests.

### Author Contributions

- Daniela E. Winkler conceived and designed the experiments, performed the experiments, analyzed the data, contributed reagents/materials/analysis tools, wrote the paper, prepared figures and/or tables, reviewed drafts of the paper.
- Thomas M. Kaiser reviewed drafts of the paper.

### Supplemental Information

Supplemental information for this article can be found online at http://dx.doi.org/10.7717/peerj.1002#supplemental-information.

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
