# Peer review of "Uneven distribution of enamel in the tooth crown of a Plains Zebra (Equus quagga)"

_PeerJ, doi:10.7717/peerj.1002_

## Round 0.1 · original submission · Major Revisions

Three reviewers have evaluated the ms and provided their comments below, and in the attached pdf. I also have some additional small comments to the paper.

Overall, all reviewers have commented positively that the paper will be a solid contribution to PeerJ following some revisions, and I am in agreement with their suggestions. Particularly, I ask if you could pay careful attention to:

1) The comments on how to improve content in the introduction and discussion (Reviewer 1). This will help the general reader and improve accessibility.

2) Points made by Reviewer 3 (Alistair Evans) regarding clarification of the methods, and suggested improvements to several figures. These must be addressed in a revision. I also find the details on how you extracted data from the models a little sparse, making it difficult for another reader to replicate – those could be simply and quickly added.

3) Comments by Reviewer 2 (Marcus Clauss) to take care with the weight of conclusions drawn given the data. This is especially important when using words such as, “constraint” without precise definition.

Other minor comments by line

Pg 2, ln27: typo, “hence expose”
Pg2, ln27: define “attritional and abrasional” prior to use
Pg2, ln62: why “at approx. 75%, 50% and 25%” and not at exactly those values using measurements?
Pg2, ln67-68: I think the reader would benefit here from some more precise detail on how you extracted volumes from the enamel sections in VGStudioMax.
Pg2, ln68: define txM1
Pg4, 84: “is getting thinner” is a little awkward, perhaps rephrase
Pg4, ln111: should be “the M3 is” also, the rest of this sentence doesn’t make sense – please check
Pg4, ln113: remove additional period
Pg5, ln119: delete “very” before “taxon”
Pg5, ln122, shouldn’t be a comma after “illustrate” also, this sentence could be rephrased for clarity – what do you determine as “severe need of optimization”? Further it is unclear how you have evaluated the significance of “basic constraints” “in terms of functional optimization” – what is a basic constraint? – how so defined?
Pg5, ln125: again, “as these constraints” what specific constraints are you referring to?
Table 1: these values are only to the nearest mm? Can you provide higher accuracy?
Fig 4 could also be enhanced significantly by use of colour and more easily distinguishable patterns, e.g. spots, stripes, rather than quite similar shades of grey.

Reviewer 1 ·

Basic reporting

No comments

Experimental design

No comments

Validity of the findings

No comments

Additional comments

Dear Editor,

In this manuscript the authors examine enamel thickness in Zebra hypsodont teeth, finding an uneven distribution, which is probably a response to mechanical masticatory stresses. The manuscript is certainly of interest to the readers of PeerJ, however it is still not ready for publication in its present form. Some important aspects need to be addressed:

1. First of all, the manuscript needs to be polished: there are several sentences not clear and difficult to understand. Moreover, there is often an awkward word choice. For example, the food is not disintegrated through mastication but it is reduced (or break down) into smaller pieces. There are also many typos that should be corrected. The authors should send their manuscript to an English native speaker for proof reading.

2. The manuscript needs a larger introduction to help a general reader to understand the importance of this study. For example, the authors do not define the difference between attritional and abrasive wear. They do not give enough details on feeding behavior in Equus quagga, and they do not provide any information on the food physical properties eaten by this species. Moreover, it would certainly help introducing some example of different prismatic structures and how they respond to the intake of different diets.

3. In the introduction will also be useful to morphologically and functionally describe the hypsodont teeth of Equus quagga. A figure that illustrates the morphology of a hypsodont tooth would be important as well.

4. The discussion needs to be significantly more robust. Based on their results, the authors need to explain the relationship between physical properties of food and enamel thickness in the different positions. Moreover, the authors should expand the part of the functional significance in having an uneven enamel distribution, taking into account the mechanical stress posited by an herbivorous diet.

5. Figure 1. The authors need to indicate in the picture the different dental tissues.

6. Figure 3B. It would be better to indicate the different sections as in the previous figure.


The are also some minor points to be addressed:

1. Page 2, line 12: the tooth positions need to be defined.

2. Page 2, line 14-15: “Newly erupted hypsodont cheek teeth share a feature between all taxa: they are not immediately functional.” Citation needed.

3. Page 2, line 15: the word “disintegrate” should change.

4. Page 2, line 22: “Every et al. 1998” needs a comma (Every et al., 1998).

5. Page 2, line 25: “and also in man”. Citation needed.

6. Page 2, line 27: The term “abrasional” is incorrect and should be changed into abrasive.

7. Page 2, line 27: Typo. "henceexpose” should be hence expose.

8. Page 2, line 29: The term “top” (the top of the crown) does not sound good because it is too vague. This should be changed into something more accurate (e.g. cusp, occlusal, coronal).

9. Page 3, lines 51-52: When citing articles published in the same year, these should be placed in alphabetical order (Fortelius and Solounias, 2000; Archer and Sanson, 2002; Solounias and Semprebon, 2002).

10. Page 3, line 78: The sentence “contained more than 9.6% more enamel”, should be changed into “contained more than 9.6% enamel”.

11. Page 4, line 89: The expression “the remainder of the tooth” is not clear.

12. Page 4, lines 97-99: The sentence “as the larger content of enamel should strengthen it and help resist high pressure and stress loads” is not clear.

13. Page 4, line 102: It is not clear what the authors mean for “the two fossettes”. The authors should define what fossettes mean (maybe in the introduction when they describe the general anatomy/morphology of a hypsodont tooth), or should use another anatomical term.

14. Page 4, lines 105-106: The sentence “this phenomenon to adaptive pressures related to generally two phenomenon” contains two times the word phenomenon.

15. Page 4, line 111: Typo. “M3is” should changed into M3 is.

16. Page 4, lines 112-113: The sentence “Therefore it is also the tooth position maintaining function when anterior teeth have already been worn out” is not clear.

17. Page 4, lines 115-117: The sentence “Because it comes in occlusion while shear-cutting functionality in anterior teeth is well established, there is no need for a weakened crown top as in other cheek teeth” is not clear.

18. Page 6, lines 178-179: The verb “separate” is used two times in the same sentence; choose a synonym.

·

Basic reporting

I made all my suggestions directly in the WORD manuscript file.

The article language is clear (except where commented), is adequately courteous and clear in expression (except where noted), presents sufficient background and prior literature (except as noted), has a correct structure, all figures are relevant and of sufficient quality, and is in total 'self-contained'

Experimental design

The submission represents primary research in terms of a pilot study.
Research question clearly defined, which is relevant and meaningful. High technical standard, sufficient information, no ethical problems.

Validity of the findings

The data represent a pilot investigation and are hence not 'statistically sound'. In my view, this is no problem if the fact that only one specimen was investigated is announced in the title (as suggested). Because of this pilot study character, the conclusions should be worded a bit more carefully (and especially not as if the finding was already representative for this species), as suggested by my comments.

Additional comments

This is a well-written pilot study, on which I have some minor comments in the attached WORD file of the manuscript (because of webpage settings, I had to convert this to pdf before attaching, which means you have to check the pdf carefully).

·

Basic reporting

No Comments

Experimental design

L55: does ‘resolution’ refer to pixel or voxel size? If so, this is a more accurate description. Also, are you sure that voxel size varies by more than a factor of 10, from 75 microns to 1000 microns? Were the teeth scanned individually or while in the skull?
L68: ‘txM1’ and similar abbreviations should be defined.
Table 1: is the precision of the measurement of enamel thickness only to the nearest millimetre? I think it would be more informative to create vertical cross-sections that show the change in thickness of the enamel vertically down the tooth for the two ridges.

Validity of the findings

Make it clear that you are measuring the proportion of enamel compared to all mineralised tissue (if that is correct).

Among of cementum deposition varies along the tooth – there is often more deposition of cementum on apical parts of the tooth compared to lower down. This therefore reduces the measured proportion of enamel in Section 1 compared to other sections. It would be better to show an analysis that excludes cementum, or show proportions of all three dental tissues separately, so we can see if an apparent increase in enamel proportion is actually due to decrease in cementum.

If the enamel close to the apex of the tooth is lower density, then it may not have been included in the enamel model. Can you be sure that the thresholding takes this into account? This is potential reason for why Section 1 appears to have less enamel than the other sections.

How does having teeth that have not mineralised their full height affect the results? Has M3 completed mineralisation and closed its roots? It would be a good idea to include an image of all 6 teeth in lateral view so the reader can get an idea of their relative size, degree of wear and eruption.

Additional comments

Abstract, L1-2: change ‘get in wear’ to ‘be partially worn’.
Abstract, last line: change ‘while’ to ‘when’.
L14: change ‘between’ to ‘among’.
L21: ‘bring the teeth in wear’ – this is unclear. Does this mean to commence wear?
L27: change to ‘hence expose’.
L62: delete comma.
L76: delete ‘even’.
L80: ‘composed of more enamel’ – that is not strictly what you are measuring. You are measuring enamel/(all mineralised tissue).
L110: ‘maintain chewing evenly distributed forces induced’ is unclear.
L111: change to ‘M3 is’.
L116-117: ‘no need for a weakened crown top as in other cheek teeth’ – this puts a completely different perspective on why the enamel proportion may be lower closer to the apex. This sentence is arguing that you need lower enamel proportion in M1 because functionality is not established, but this is not the case for M3. I do not understand how this follows.
L124: ‘basic constraints are rendered insignificant’ – unclear. Please simplify.

Fig. 1: it would be useful to show the extent of the ‘mineralised’ tissues on A by colouring the non-mineralised tissue another colour. This would retain the contrast between enamel, dentine and cementum while allowing the reader to see the extent of the thresholding that includes mineralised tissue. Another idea would be to keep A as it is, but then in B show thresholded ‘all mineralised tissue’ in one colour and ‘enamel’ in another.

Fig. 2: while the model looks very nice, it is a bit too dark to clearly see aspects such as the root structure.

Fig. 4: this does not clearly show the differences among teeth of the different sections. It may be better to show all sections as separate bars from the x axis. This would enable the enamel proportion to be compared both within a tooth and among teeth. The only information that would be lost would be the average enamel proportion per tooth (as represented by the figure at the top of the stacked bar chart), which could be included elsewhere.

---

## Round 0.2 · Minor Revisions

I thank the authors for carefully attending to the earlier comments and revising their manuscript accordingly. Both reviewers are in agreement that this version is much improved and almost ready for publication. I agree with their comments and have very few minor points to add beyond their helpful suggestions. Particularly, R1 has noted that the authors need to be careful with their use of the terms abrasion and attrition throughout the manuscript, and provides some suggestions to improve the captions for Figures 1 and 2 (see also the suggestions of R2). The suggestions of R1 will make the results more accessible to the reader, and I encourage the authors to address those points.
R2 has listed several points that require clarification, particularly please state whether the roots are closed in the teeth examined for this study, and please address the points raised by R2 with regard to the orientation of the teeth (Figs. 1 and 3) and the segmentation of the pulp cavity (Fig. 2). Additionally, I agree with R2 that detailing the raw volumes for the dental tissues would be more accessible for the reader.
Finally, please take care to be consistent with the use of British/American spelling and use of “apex” or “top” throughout.

Minor points
Ln11: I suggest, “more simply” rather than “simpler”
Ln34: delete apostrophe
Ln35: perhaps rephrase this sentence, “Grass lamina is generally narrower and more uniform in thickness than browse” or similar
Ln63: should be “voxels”
Ln116: delete “a” after “by”
Ln118: replace “at a” with “to a”
Caption for Figure 3: should read “cross section through”
Use of the word “ratio” on ln87 and caption for Figure 4. A ratio is expressed for example as 4:3 or 3:4, and I think the values reported in Figure 4B do not reflect that notation. I would consider removing the word “ratio” and use either “value” or “amount”
Reference to “Tab.” should be “Table” and please check “Section” and “Stage”, which are presently used interchangeably in the text/table/figures.

Reviewer 1 ·

Basic reporting

The revised manuscript has significantly improved since last version, however, I do believe that there is still some work to do prior publication.

Experimental design

No comments

Validity of the findings

No comments

Additional comments

One of the major problem of the earliest version was the lack of clarity and the wrong words choice used. Unfortunately this problem still persists in the revised manuscript: some parts are still unclear and difficult to follow.

Some terms, like “food disintegration” should be removed completely from text. Chewing reduces food into small pieces providing area for the digestive enzymes. The absorption of small food molecules is completed in the digestive system. The food is not completely "disintegrated" with chewing, which is only the initial phase of digestion and its final part.

Moreover, the authors should use either British English or American English and not both. Finally there are still some grammatical mistakes that need to be corrected.

The terms abrasion and attrition are still poorly described and incorrect. For example, abrasion does not refer only to the food-tooth contact but also to the external small particles present in the environment, such as grit and dust.

The introductory paragraph about thegosis is unclear.

Figure 1 should better describe what are the key elements of an hypsodont tooth. Because the whole manuscript focuses on hypsodont teeth, this should be better described in the text and in the figures. It is certainly not enough from the authors to say that most readers of the scientific community do know what a hypsodont tooth looks like. There are many readers from different scientific fields that do read manuscript from other disciplines.

Figure 2: the authors needs to point in the figure the different dental tissues. For example, in Figure 2 B because there are different browns, is not clear at all where cementum is.

Minor points:

1. Line 25: The authors need to cite the source of information related to thegosis/bruxism in man;

2. Line 42: While beginning the text with "Specialized" when the previous part was written in British English?

3. Lines 54-55: When citing more articles published in the same year, these should be ordered in an alphabetical order.

4. Lines 99-100: The sentence should be re-phrased. The authors used the same term (phenomenon) twice in the same sentence.

5. Lines 105-106: Sentence unclear: re-phrase it.

6. Lines 111-113: Sentence unclear: re-phrase it.

·

Basic reporting

No Comments

Experimental design

No Comments

Validity of the findings

No Comments

Additional comments

This is a substantial improvement on the original submission, but there remain or have been added a few points of confusion that should be clarified before publication.

Abstract L5: ‘upper half of the tooth crown in all upper tooth positions’ – it would be clearer if the first ‘upper’ were changed to ‘apical’ or similar; likewise through the rest of the manuscript.
Abstract L7: ‘promote early wear’ – the use of the word ‘promote’ implies that this is a desired consequence, i.e. that there is an advantage to having early wear in the apical half of the tooth crown. Suggest you reword this if it is not intended.
Abstract L7: change to ‘Besides’.
Abstract L9: ‘different functional trait in mastication’ – perhaps ‘different functional consequences in mastication’ instead.
L34: Delete apostrophe after “Its”.
L35: ‘Grass lamina width is generally less’ – is this to be associated with ‘uniform’ later in the sentence? It is unclear – please restructure sentence.
L39: delete full stop after ‘grinding’.
L52: if the teeth have been removed from the skull then they are not ‘articulated’.
L61: change ‘recreated’ to ‘reconstructed’.
L63: change to ‘voxels’.
L69: insert ‘the apex at’ before ‘100’.
L96: change ‘fast’ to ‘faster’ – it is still slower than almost all other mammals.
L100: delete ‘generally’ or move it elsewhere in the sentence.
L118: change ‘at’ to ‘to’.
L118: ‘even hypsodont forms are prone to high levels of functional adaptation’ – I don’t think this is necessary. The mere fact they are hypsodont shows a high level of functional adaptation.

There should be some definitive statement about whether the roots are closed on any or all of these teeth – is enamel deposition continuing for any or all of these? This is important has it gives a better indication of how much of the tooth has not been included in this analysis.

The manuscript switches between ‘apex’ and ‘top’. I think the former is better, because when dealing with ‘upper’ dentition, when placed in life orientation, the ‘top’ of the tooth is the root.

Fig. 1: ‘Anatomical position’ is vague. You mean that the teeth are positioned relative to one another as they would be in life? You should indicate the view of A. It is rather hard to see on the surface models in B, but I think P2 does show occlusal wear (more effective shadowing would make this clearer). If this is the case, then the positioning in A is not correct – P2 should be further down the page so that it could line up with the occlusal plane of M1 and M2. I can’t see the surfaces of P3 and P4 clearly enough to know whether they have wear. B is also more occlusal-lingual view.

Fig. 2: in B it seems to indicate that the pulp cavity (with black voxels and low density) has been segmented to ‘dentin’. Surely this is not the case? This goes for the cracks in the tooth as well. Also, there is some difference in the colouring or representation of the dental materials between A and B such that if you removed the colours they are not the same picture. Perhaps B has some amount of surface rendering? Either way, the dental material in B should be identical to A with only the segmentation colours added

Fig. 3: Is the orientation of the teeth different between figure parts? Apical appears to be at the top of the picture in A but bottom in B and C. Orientation should be defined. Why was P3 chosen as the tooth to label the parts when A and B show M1? Also, the colouring showing the different sections should be a lot clearer. Was the base of M1 cut at the cementoenamel junction in this figure?

Fig. 4: The term ‘Stage’ is used here while other parts of the manuscript refer to ‘Section’. Please be consistent.

Table 1: I think it would be more useful here to report the actual volumes (in mm^3) of the three dental tissues rather than repeating some of the same data that is in the figures. This way, readers can examine for themselves how absolute and relative amounts change. If you want to include the proportional values as well that’s fine, but I think it’s not necessary, and the volume data is more important.

---

## Round 0.3 · accepted · Accept

Thank you for carefully revising the manuscript in line with comments of the reviewers. I think this version is much improved, and now ready to be published.

I suggest three very minor editorial points that should be quickly rectified in the final version:

Ln25: I would replace the last part of this sentence, “and in lack of empirical data..” with “and a lack of empirical data are available in support of this hypothesis”
Ln106: replace “phenomenon” with “result”
Ln115: parentheses are missing for Kaiser 2002